# New deep learning method for efficient extraction of small water from remote sensing images

Yuanjiang Luo[1], Ao Feng[1], Hongxiang Li[1], Danyang Li[1], Xuan Wu[1], Jie Liao[1], Chengwu Zhang[2], Xingqiang Zheng[3], Haibo Pu[1]*

1 College of Information Engineering, Sichuan Agricultural University, Ya'an, Sichuan, China, 2 College of Information Science and Technology, Nanjing Forestry University, Nanjing, Jiangsu, China, 3 College of Management, Sichuan Agricultural University, Chengdu, Sichuan, China

☯ These authors contributed equally to this work.
* puhb@sicau.edu.cn

**Data Availability Statement:** All relevant data are within the paper and its Supporting information files.

**Funding:** The author(s) received no specific funding for this work.

## Abstract

Extracting water bodies from remote sensing images is important in many fields, such as in water resources information acquisition and analysis. Conventional methods of water body extraction enhance the differences between water bodies and other interfering water bodies to improve the accuracy of water body boundary extraction. Multiple methods must be used alternately to extract water body boundaries more accurately. Water body extraction methods combined with neural networks struggle to improve the extraction accuracy of fine water bodies while ensuring an overall extraction effect. In this study, false color processing and a generative adversarial network (GAN) were added to reconstruct remote sensing images and enhance the features of tiny water bodies. In addition, a multi-scale input strategy was designed to reduce the training cost. We input the processed data into a new water body extraction method based on strip pooling for remote sensing images, which is an improvement of DeepLabv3+. Strip pooling was introduced in the DeepLabv3+ network to better extract water bodies with a discrete distribution at long distances using different strip kernels. The experiments and tests show that the proposed method can improve the accuracy of water body extraction and is effective in fine water body extraction. Compared with seven other traditional remote sensing water body extraction methods and deep learning semantic segmentation methods, the prediction accuracy of the proposed method reaches 94.72%. In summary, the proposed method performs water body extraction better than existing methods.

## Introduction

The accurate acquisition of information on the distribution of surface water bodies is significant in the fields of water resources investigation, comprehensive river management, water resource planning, flood and drought monitoring, and disaster assessment [1]. With the increasing number of artificial earth satellites, abundant and detailed satellite remote sensing

**Competing interests:** The authors have declared that no competing interests exist.

image resources are becoming increasingly available. Rapid and accurate extraction of water body information from satellite remote sensing images has become an important tool for surface water resource investigation and monitoring.

Several water body extraction methods have been proposed. Traditional methods involving automatically extracting water body information using remote sensing technology include the spectral classification method [2], single-band threshold method [3], and water body index method [4]. The spectral classification method classifies water bodies from background features based on the differences of spectral features from the images and then extracts water body information. Water discrimination has been proposed to improve by fusing spectral indices [5]. The single-band threshold method exploits the strong absorption of water bodies at the near- or mid-infrared band and selects the maximum value that makes the reflectance variance between water and non-water bodies as the threshold to extract water body information. Lu et al. used the near-infrared band to reduce the influence of artificial building sites for water body mapping [6]. The classification results of this method on remote sensing images with more shadows are unsatisfactory, directly leading into the area of water bodies extracted by the classification being significantly more than the actual area. The water body index method is widely used by researchers worldwide. The conventional indices for extracting water body information include Normalized Difference Water Index (NDWI) and Modified Normalized Difference Water Index (MNDWI). The NDWI takes advantage of the unique reflectivity of water bodies in the green band which is higher than that in the near infrared band, and normalizes the difference between these two bands to highlight the water body information and distinguish it from its background features. This method is simple and easy to operate but also easily confuses construction land with water bodies [7]. Hanqiu analyzed the NDWI and modified the near-infrared band in the NDWI model equation to the mid-infrared band (MNDWI), whose water body extraction effect is better than NDWI by extracting urban water bodies more effectively and eliminating the influence of partial shadows on water bodies [8]. Gu et al. proposed a water body extraction algorithm for multispectral remote sensing images based on region similarity and boundary information, combining adaptive spectral band selection and over-segmentation [9]. Wang proposed a method called Remote Sensing Stream Burning (RSSB), which combines high-resolution observed stream locations with rough topography to improve water extraction and reduce the effects caused by observed data and model resolution [10]. Li et al. improved the normalized difference water index (MNDWI) and proposed the contrast difference water index (CDWI) and shaded difference water index (SDWI) to solve the water leakage problem in shaded and unshaded areas in urban districts [11].

All these studies focused on enhancing differences between water bodies and other disturbed water bodies and on improving the accuracy of water body boundary extraction. Indeed, an optimal method is unavailable; only the most suitable method is used for a target study area. Owing to the problems of mountain shadow obscuration, shallow water disconnection, and high transparency of some water bodies in reality, a combination of methods is required to extract water body boundaries more accurately.

With the development of artificial intelligence technology, applying deep learning to information extraction in the remote sensing field has become a hot topic for researchers. Some researchers have applied semantic segmentation to remote sensing image interpretation and achieved good results [12, 13], such as automatic mapping method of urban green spaces (UGS) [14] and novel spatiotemporal neural network [15]. Recently, deep learning has been increasingly applied to the extraction of water body information from remote sensing images. Qi et al. combined convolutional neural network (CNN) with Markov model and used semi-supervised learning strategy to reduce data dependency improving the extraction performance

of global and local water bodies by 7–10% [16]. Chen et al. developed a global spatial-spectral convolution and surface water body boundary refinement module to enhance surface water body features. They also designed the WEB-NN architecture to segment high-resolution remote sensing images [17]. Wang et al. applied a full convolutional network (FCN) to extract lake water bodies from Google remote sensing images [18]. Zeng et al. proposed a FCN with the RCSA mechanism [19] for the large-scale extraction of aquaculture ponds from high spatial resolution remote sensing images. This study proposed a CNN-based framework to recognize global reservoirs from Landsat 8 images [20]. In this paper, a new semantic segmentation CNN called the multi-scale water extraction convolutional neural network is proposed for automatically extracting water bodies from GaoFen-1 (GF-1) remote sensing images [21]. This study developed a novel self-attention capsule feature pyramid network (SA-CapsFPN) to extract water bodies from remote sensing images [22]. By designing a deep capsule feature pyramid architecture, the SA-CapsFPN can extract and fuse multi-level and multiscale high-order capsule features [23, 24]. However, these methods are quite dependent on convolutional feature extraction. In the case of complex geographic information interference, similar continuous spatial information can negatively affect the water body extraction task and thus the overall accuracy. In urban water body extraction, farmland cannot be accurately distinguished from water bodies.

In nearly all studies, existing methods struggle to improve the extraction accuracy of fine water bodies, such as urban rivers, while ensuring the overall extraction effect [20–23]. Therefore, how to find a better model suitable for high accuracy water body extraction in universal scenarios is a current priority.

In summary, there are two main challenges in water body extraction from remote sensing at the current stage:

1. In a remote sensing space, accurately extracting fine water bodies is difficult under the influence of mixed image elements while ensuring the good overall extraction effect of large-scale remote sensing images;

2. In the case of complex geographic information interference, the remote sensing image part that is highly similar to the water body negatively affects feature extraction and thereby affect the extraction accuracy.

At present, a fixed definition of fine water bodies on remote sensing images remains undefined. Jiang et al. defined fine water bodies as narrow water bodies with an apparent width of image elements less than or equal to three elements in the image [24]. In this study, a small water body is defined as a small river or pond with an apparent width of less than or equal to 15 pixels in the image. To make the algorithm applicable to both fine and large water bodies, this study does not distinguish between fine water bodies and other water bodies when testing and evaluating the algorithm, but evaluates the water body extraction results in general.

To address the existing challenges, we propose method based on deep learning for extracting water bodies from remote sensing images. The original image is processed by a GAN model to enhance the features of fine water bodies such that the network can focus on fine water during training. In addition, fine water bodies such as ponds are often far from rivers, and to better capture the remote relationships in isolated regions, this study adopts a bar-pooling method such that the scene resolution network can aggregate both global and local contexts. Rather than the disordered spatial pyramid pooling (SPP) in DeepLabv3+, a hybrid pooling module (MPM) is used to detect complex scene images using different core shapes. These improvements allow our model to perform water body extraction better than existing methods.

Our contributions are summarized as follows:

1. We propose a new deep learning-based water body extraction method for remote sensing images, which reconstructs the images to enhance fine water body features.

2. We introduce a bar pool using detailed qualitative and quantitative evaluation to demonstrate the advantages of our method concerning water body extraction.

3. We propose a strategy that enables multi-scale input while lowering the training cost.

In the remainder of this paper, we first briefly introduce the various methods used in this study. Then, we introduce our data sources. Finally, we detail the methods proposed in this study, conduct experiments, and conclude the paper.

## Related work

### DeepLab

DeepLabv3+ is the latest algorithm in the DeepLab family, a variant of DeepLabv1 [25] and DeepLabv2 [26]. DeepLabv1 first mentioned dilated convolution, which solves the multi-scale problem of semantic segmentation. DeepLabv2 adds ASPP to DeepLabv1 to solve the multi-scale problem by inputting a feature map into multiple dilated convolutions with different expansion rates (Fig 1). The resulting feature maps are fused and then upsampled. The module designed by Deeplabv3 [27] performs Atlas convolution in a cascaded or parallel manner to capture the multiscale context by employing multiple Atlas rates. In DeepLabv3, the final generated feature maps directly output the prediction results after 8- or 16-fold upsampling. DeepLabv3+ [28] fuses the feature maps output using the ASPP module with one of the layers in the CNN and upsamples them to obtain the final prediction results. DeepLabv3+ can better fuse the high and low level features and retain both boundary and semantic information. In addition, the fusion of multi-scale information is performed by an encoder-decoder, while preserving the dilated convolution and ASPP layers used in the previous series. The backbone network utilizes an improved Xception model with different perceptual fields and upsampling to achieve multi-scale feature extraction and uses depth-separable convolution to reduce the number of parameters.

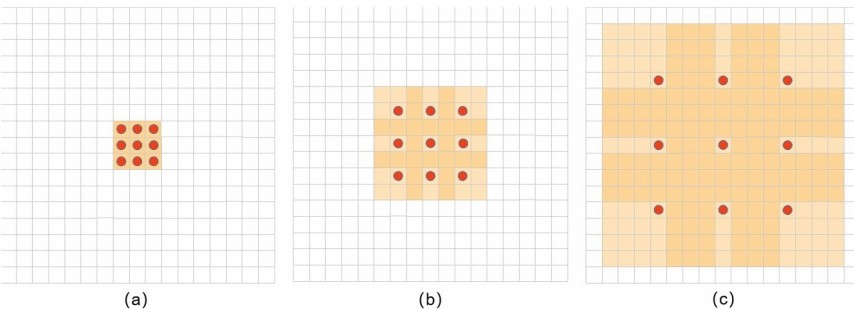

**Fig 1. Improved DeepLab network.** (a) is an ordinary 3 × 3 convolution kernel, which can also be understood as the void convolution with dilation rate = 1 and is a special form of the void convolution. (b) is a void convolution with a dilation rate = 2. Based on the ordinary 3 × 3 convolution kernel, it expands a 3 × 3 convolution kernel into a 7 × 7 convolution kernel by adding empty points with a weight of around nine points, thereby increasing the receiver field. (c) Similarly to (b), it has a dilation rate = 4 and expands the 3 × 3 convolution kernel to 15 × 15.

## Generative adversarial networks (GANs)

GANs train two models simultaneously [29] a generator network (G) that captures the data distribution and a discriminator network (D) that estimates sample probabilities from the training data. The training task of G is to maximize the probability that D makes an error. This framework allows us to prove that a unique solution exists in the space of arbitrary functions G and D such that G reproduces the training data distribution. In the case where G and D are defined by a multilayer perceptron, the entire system is trained using backpropagation. Markov chains or extended approximate inference networks are not required for the training or sample generation process. In the discriminative model, the loss function is easily defined owing to the relative simplicity of the output target. However, the definition of the loss function for generator networks is relatively complex. The expectation value of the result is often a vague paradigm that is difficult to define axiomatically. Thus, the feedback part of the generative model is assumed by the discriminative model. The potential of the framework is evaluated qualitatively and quantitatively on the generated samples. In recent years, many researchers have used GANs for image generation [30] with data enhancement. Xi used DRL-GAN to enhance tiny object features from very low resolution UAV remote sensing images and extract them [31]. The objective function of GAN can be defined with Eq (1)

$$\min_{G}\max_{D}V(D, G) = E_{x \sim P_{data}(x)}[lgD(x)] + E_{z \sim P_z(z)}[lg(1 - D(G(z)))] \tag{1}$$

where z is random noise and x denotes the real data.

## Strip pooling

Hou et al. proposed a new pooling strategy that reconsiders the form of spatial pooling and introduces a strategy called strip pooling [32]. This strategy uses a long and narrow core (i.e., $1 \times N$ or $N \times 1$) and proposes two pooling-based network modules for scene analysis. The strip pooling module (SPM) can effectively expand the receptive field of the backbone network. The SPM consists of two paths that encode contextual information primarily along the horizontal or vertical spatial dimension. For each spatial location in the feature map generated by pooling, the module encodes its global horizontal and vertical information and then uses these encodings to balance its weights for feature optimization (Fig 2).

The pyramid pooling module (PPM) is an effective manner of enhancing scenario analysis networks. Although the pyramid has different pooling kernels, the PPM primarily relies on standard spatial pooling operations. Considering the advantages of standard spatial and strip poolings, Hou et al. improved the PPM by designing the MPM, which is dedicated to summarizing different types of contextual information through various pooling operations to further

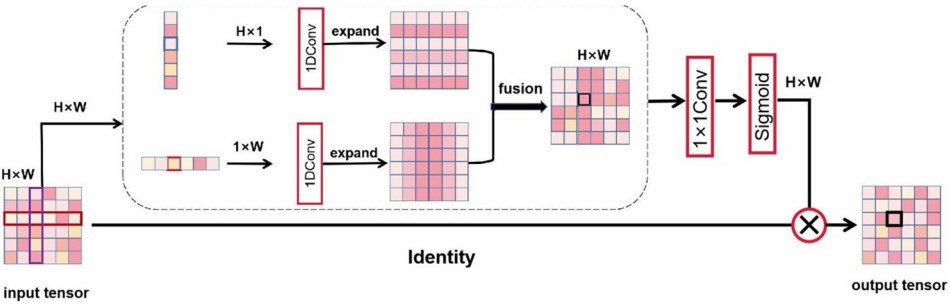

**Fig 2. SPM structure diagram.**

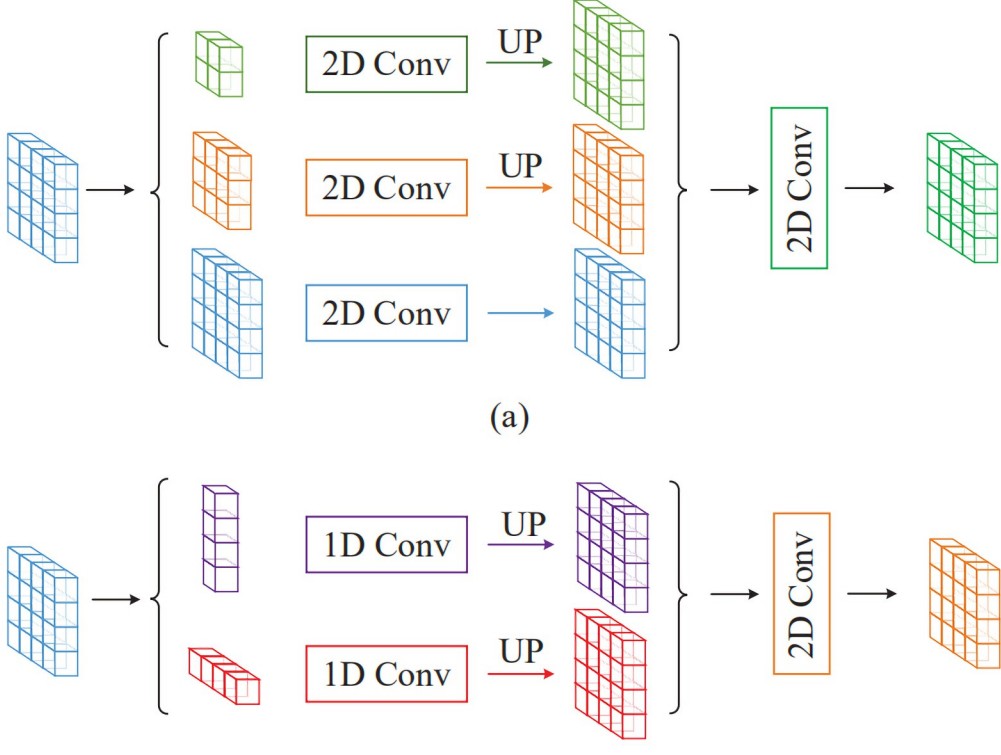

**Fig 3. MPM structure diagram.**

differentiate feature representation. The MPM utilizes novel additional residual building blocks to model remote dependencies at a high semantic level. Using pooling operations with different kernel shapes to detect images with complex scenes, contextual information can be collected in a complete manner (Fig 3).

## False color processing

False color synthesis, also known as color synthesis, is based on the additive or subtractive color method. The multi-segment monochrome image synthesis of a false color image is a special color enhancement technique. Synthetic color image is different from natural color and can be transformed arbitrarily; thus, it is called false color image. Remote sensing images are sensitive to texture and color [33]. The remote sensing images of Sentinel-2A provide a variety of band data such as concerning the panchromatic, near-infrared, and green bands. The green, red, and infrared bands from remote sensing data are assigned to the blue, green, and red bands in RGB, respectively, which can be converted to standard false color images. In unprocessed remote sensing images, the colors of vegetation and water bodies are similar, but vegetation turns red in false color images while the water bodies turn green, blue, dark blue, etc. depending on the number of microorganisms contained.

## Materials and methods

### Dataset

The Sentinel-2A satellite is the second satellite of the GMES program and provides a unique global perspective [34]. Sentinel-2A was launched on June 23, 2015, and carries a multispectral

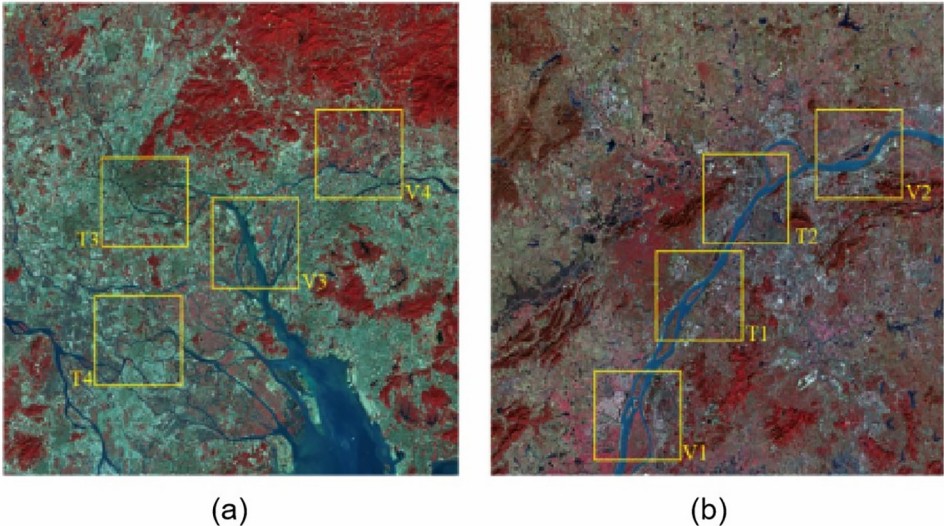

**Fig 4. Images of the Yangtze River basin and the Pearl River Delta region.** The picture shows the remote sensing images of (a)Nanjing and (b)Guangzhou, China, processed by standard false-color synthesis, where regions T1-T4 and V1-V4 are divided into training sets and test sets, respectively. Reprinted from www.gscloud.cn under a CC BY license, with permission from Dr. Qinghui Lin, original copyright 2019.

imager covering 13 spectral bands with a width of 290 km. Sentinel-2A satellite data are available from the European Space Agency's Sentinel online platform, with a spatial resolution of 10 m and revisit period of 10 days. The short revisit time is convenient for continuous acquisition and water information monitoring. Once a robust prediction model is established, real-time extraction and dynamic monitoring of water in remote sensing images can be achieved.

We used the Sentinel-2A satellite to acquire images of the Yangtze River basin and Pearl River Delta region in China on December 11 and December 28, 2019. After processing the false color (Fig 4) as the study object, and with the aid of colleagues concerning remote sensing and computer vision, we manually annotated the images, ensuring a proper division of water and non-water parts.

As the size of the remote sensing images from Sentinel-2A are approximately 10,000 × 10,000 pixels, manually labeling and batch training the entire image simultaneously is costly. Therefore, we trained and predicted the cut remote sensing images, and output the prediction results of the entire image using the window sliding strategy by stitching overlapping steps.

## Data preprocessing

As the proportion of tiny water bodies to the entire space is relatively small in the large-scale remote sensing space, detecting tiny water bodies is difficult. In addition, the low contrast of the unprocessed remote sensing image, which is affected by mixed image elements, makes extracting tiny water bodies difficult.

To enhance image contrast, we use false color processing. Among the various bands, the near-red and red bands are sensitive to water bodies. In addition, in the original remote sensing image, the vegetation is highly similar to the water body, and we used the green band to enhance the contrast between the vegetation and water body. Hence, vegetation will be predominantly red, and the water body will be green, blue, dark blue, etc. depending on the number of microorganisms contained. The contrast between the vegetation and water body is enhanced while minimizing the change in the characteristics of the water body. In summary,

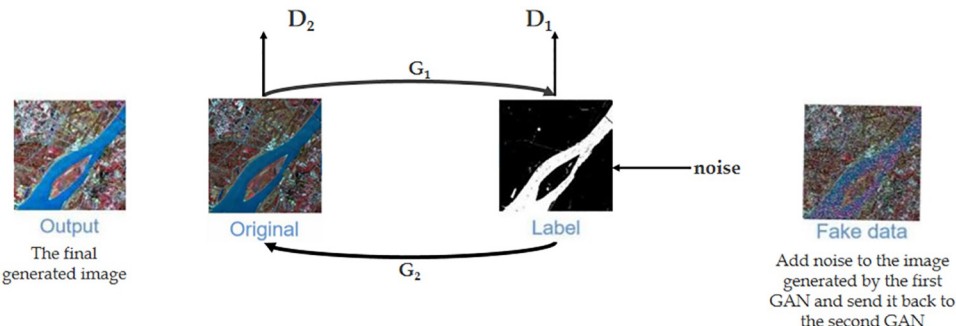

**Fig 5. Data enhancement with two generative adversarial networks.** The process of forming a loop with two GAN networks to generate images. Reprinted from www.gscloud.cn under a CC BY license, with permission from Dr. Qinghui Lin, original copyright 2019.

we adopted the standard pseudo-color processing scheme involving the assignment of the green, red, and infrared bands of remote sensing data to the blue, green, and red bands of RGB, respectively. The NIR band used is located in the highly reflective vegetation zones, reflecting plant information, and in the strong absorption zones of water bodies, enabling the identification of water-related geological formations and outlining water body boundaries. The green and red bands further highlight the distinctions between water and vegetation and help improve the accuracy of water extraction. The experiment results show that the overall detection effect improved using standard false color processing.

To enhance small water body features, we trained a generator network that can accurately reinforce these features using GANs. Throughout the process, in addition to building the network model, we manually labeled numerous remotely sensed images of small water bodies. Discriminations were made by a standard discriminator network, and after continuous adversarial training, a generator network capable of accurately enhancing the features of tiny water bodies was obtained and incorporated into the subsequent improved DeepLabv3+ network as a predecessor network. As the initial data in the original GAN is random noise and the network only requires the generated image of the generator to approximate the real image without setting constraints on its content, the generated image may not match our expected content despite its realism. To make the generated images fit our expected content as much as possible, we used two GANs in a cyclic manner to form the network, whose structure is shown in Fig 5. In our GAN, we input the original image into the first GAN, use its generator $G_1$ to generate images, and subsequently input the generated images into its discriminator $D_1$ to discriminate whether the generated image of $G_1$ is true according to the label. Then, the generated image is fed to generator $G_2$ of the second GAN, and the generated image of $G_2$ is given to the discriminator $D_2$ of the second GAN to discriminate whether it approximates the original input image. In this manner, we obtain a generated image that is realistic to the label and retains the content of the original input image, enhancing the fine water body features in the image. In addition, to converge the imbalance between the generator and discriminator, we added artificial noise data to the output images of generator $G_1$. Fig 6 shows the original, the falsely colored and processed, and GAN-enhanced images.

## Input processing

The representativeness of training data is often more important than the quantity of data. We selected four representative regions ($2048 \times 2048$) in two $10980 \times 10480$ remote sensing images as the training set for visual interpretation and data annotation. As shown in Fig 4, the

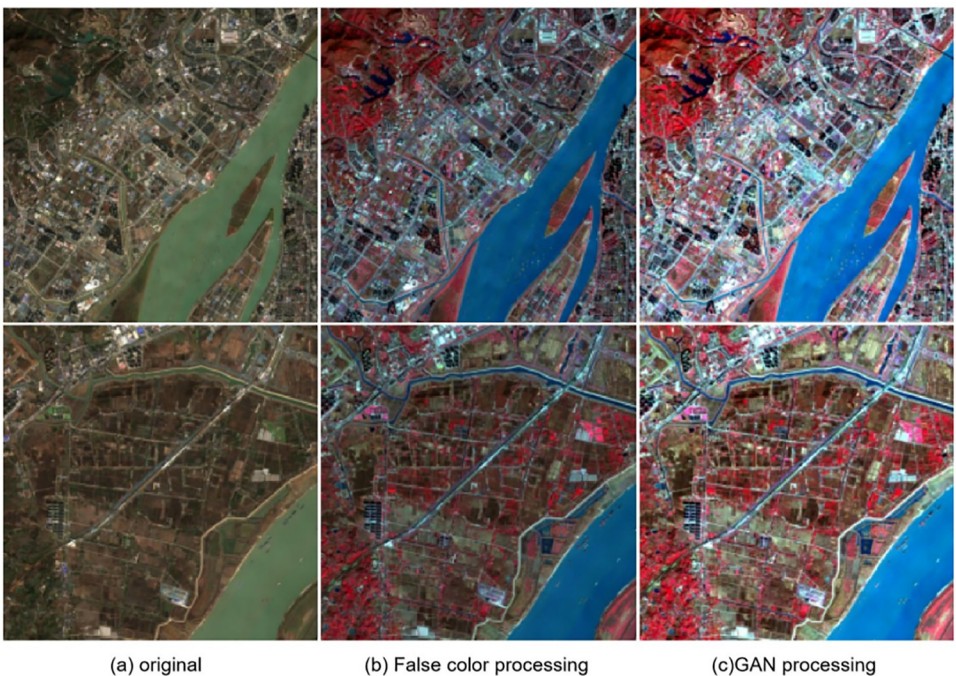

(a) original (b) False color processing (c)GAN processing

**Fig 6. Comparison of false color processing and GAN processing.** From left to right are the original image, the image after false color processing and the image after GAN processing. It is not difficult to see that compared with the other two kinds of data, the data processed by GAN has stronger contrast and more obvious water features. Reprinted from www.gscloud.cn under a CC BY license, with permission from Dr. Qinghui Lin, original copyright 2019.

selected regions T1 and T2 are agricultural fields, which are a few urban buildings and areas with penetrating water, respectively, whereas regions T3 and T4 contain many urban buildings and small rivers, respectively. After complementarily processing the features generated by the adversarial network, we labeled the data and divided the water body regions and non-water body regions as the original data for training. Then, regions V1-V4 were selected as the production validation set in these two maps.

It has been shown that the generalization performance of models with a single input size is poor. The Sentinel-2A satellite is the second satellite of the GMES program and provides a unique global perspective [35]. A larger input size of the input loses some image detail information, whereas a smaller input size generates a large amount of error owing to the complexity of the information contained in the remote sensing image which affects the final accuracy of the model. Both of these factors negatively affect the accuracy of the model to different degrees. The Sentinel-2A satellite remote sensing image data acquired in this study had a spatial resolution of 10m, but a river in the city is narrow and can be as small as 1 pixel wide in the image perception, and the width of the river crossing the city is much larger. The problem of extracting large water bodies while considering narrow rivers must be solved. The multi-scale input can train the model to accurately extract spatial information from images of different sizes, accounting for both local and global information to achieve good results in extracting large-area waters and small water bodies. Our multi-scale strategy reduces the training time cost while achieving the same training results.

To improve the extraction accuracy of tiny water bodies without losing that of large water bodies, we use neural networks with a multi-scale feature extraction strategy. In a study of multi-scale feature networks, common multi-scale feature extraction exists in the network

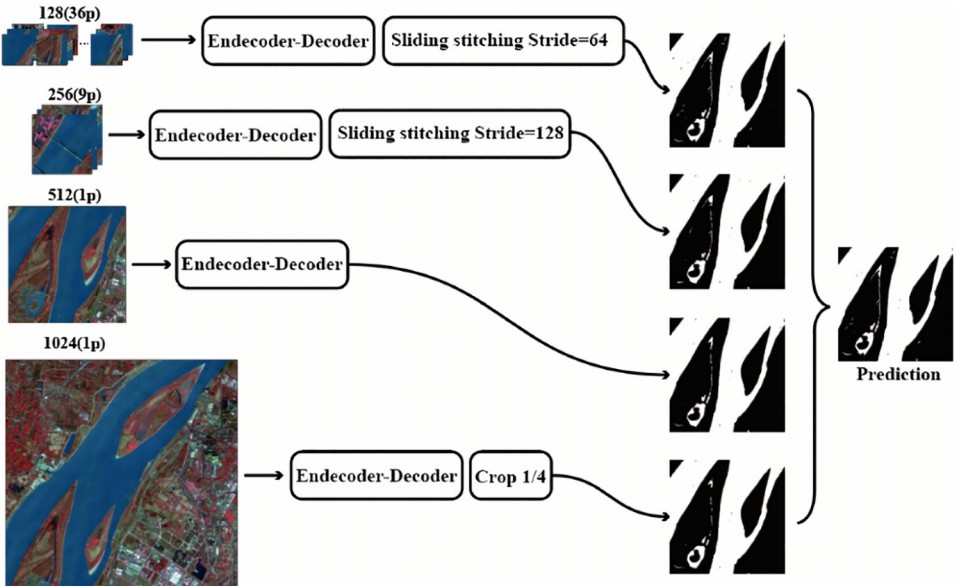

**Fig 7. The data input of various sizes is processed and then spliced into a uniform size label diagram.** Reprinted from www.gscloud.cn under a CC BY license, with permission from Dr. Qinghui Lin, original copyright 2019.

structure rather than in the data input [36]. We chose four 2048 × 2048 images for data enhancement. The neural network is sensitive to data with different orientations, different colors, and data points without noise. Therefore, in this study, we expand each image into a set of photos containing images sized 128 × 128, 256 × 256, 512 × 512, and 1024 × 1024 by randomly cutting, rotating, and adding noisy data points (Fig 7).

Then, for the 128 × 128 data map, we set the threshold value of the category proportion to 90%. The images whose proportion in one category exceeds the threshold are deleted to form the training set (Table 1).

On the input side of the model, the training images of different sizes are uniformly scaled to the same size before model training. To diminish the image scaling effect, the interpolation algorithm was used to process the images. By comparing various interpolation algorithms, we found that the Lanczos method [37] can obtain the most continuous pixel distribution for image interpolation and shrinkage. The differences between adjacent pixels can be smoothed, avoiding the deviation of eigenvalues when the image undergoes convolution (Fig 8).

In the coordinate diagram of the results of each algorithm, the abscissa represents the pixel value, and the ordinate represents the gray value of the pixel. When we scaled the data to 512, we observed that the LANCZOS algorithm performed the gentlest pixel changes between adjacent regions and had the smallest differences between pixels, making the final image look more natural.

**Table 1. Number of training samples for different image sizes.**

| Image size | Number |
|---|---|
| 128*128 | 4096 |
| 256*256 | 1024 |
| 512*512 | 256 |
| 1024*1024 | 64 |

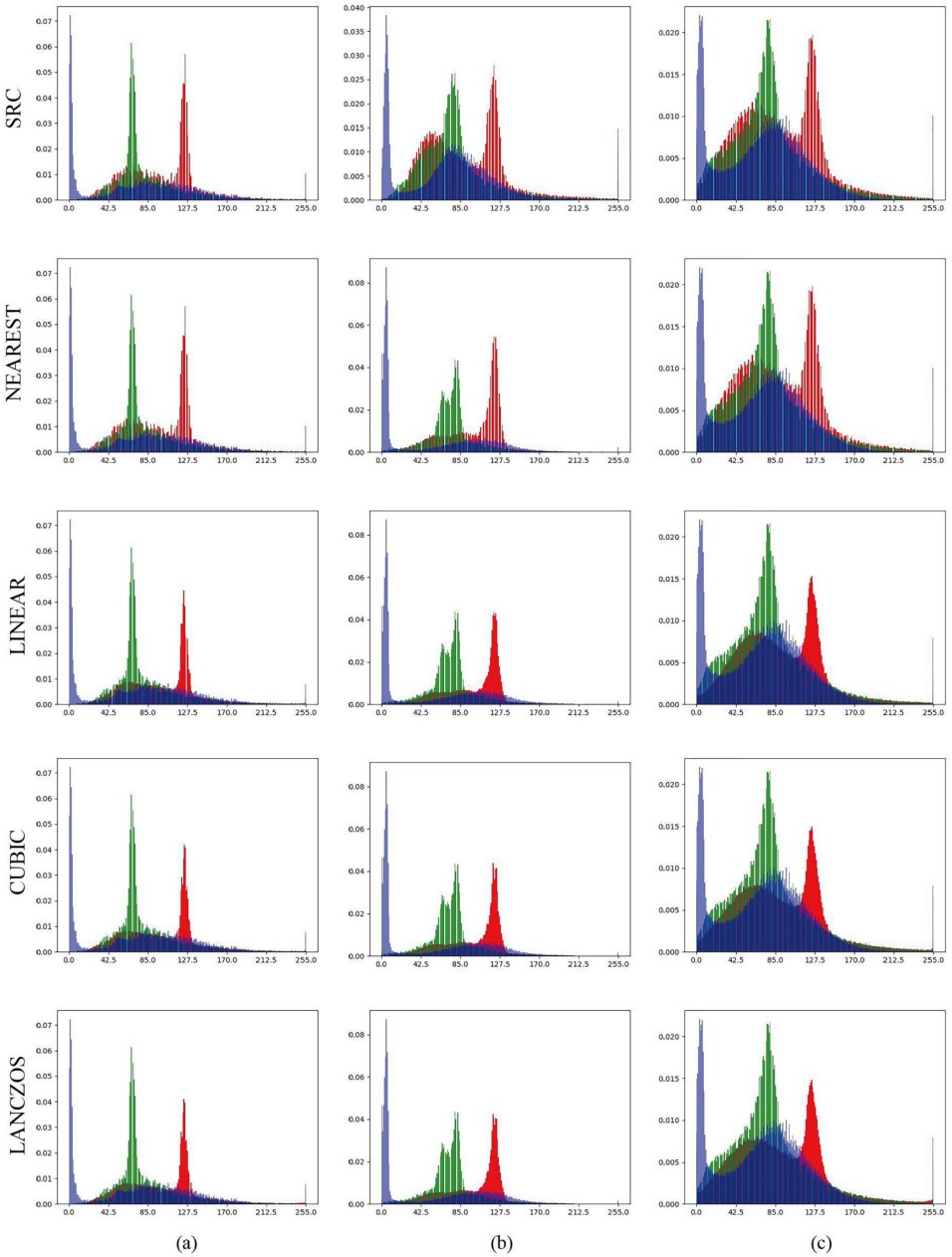

**Fig 8. Comparison of several image scaling processing methods.** Columns (**a**), (**b**), and (**c**) show the pixel distribution statistics of $128 \times 128$, $256 \times 256$, and $1024 \times 1024$ images, respectively, enlarged or reduced to $512 \times 512$. NEAREST: nearest neighbor interpolation. LINEAR: bilinear interpolation. CUBIC: bicubic interpolation of a 4x4 pixel neighborhood. LANCZOS: Lanczos interpolation of a 8x8 pixel neighborhood.

## Improved DeepLabv3+ based on strip pooling

DeepLabv3+ extracts feature information via dilated convolution. Dilated convolution extends the reception range of convolution and does not require additional parameters. However, its use of square pooling kernels limits their flexibility in capturing the contextual anisotropy, which is widely present in realistic scenes [32]. When extracting water bodies with a discrete distribution over long distances, using a large square pooling window inevitably merges

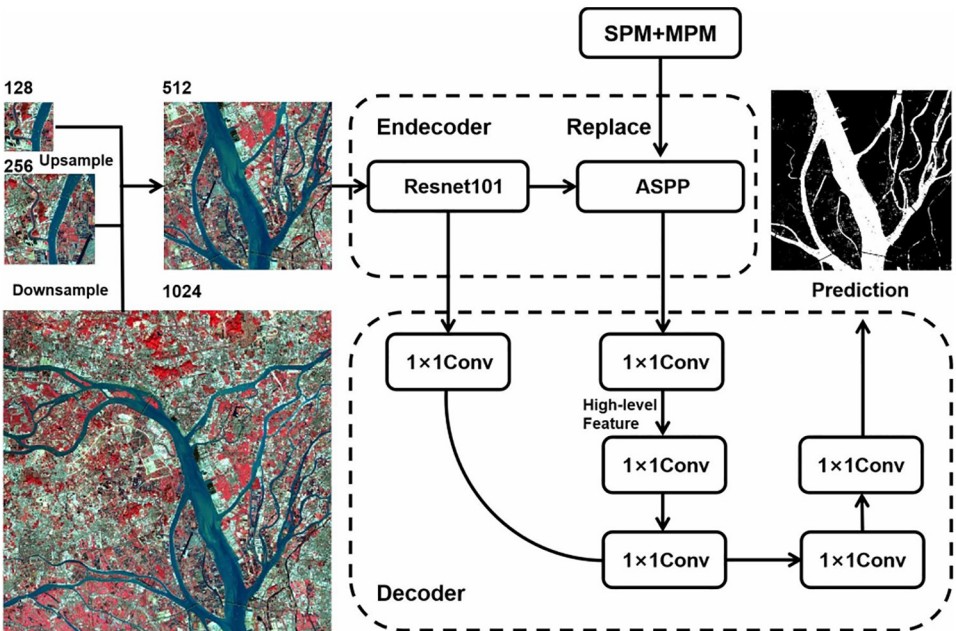

**Fig 9. Replacing the ASPP module with the SPM and MPM modules.** Reprinted from www.gscloud.cn under a CC BY license, with permission from Dr. Qinghui Lin, original copyright 2019.

contaminated information from unrelated regions and does not solve the problem effectively. In contrast, the strip pooling strategy uses a long and narrow core that effectively expands the receptive field of the backbone network and solves such problems. We uniformly scale the image size down to 512x512 at the training input side and apply strip pooling to DeepLabv3+ with SPM and MPM instead of ASPP (Fig 9). In the actual training process, the information obtained from different types of contexts can be aggregated through various pooling operations, making the feature representation more distinguishable and achieving better results in subsequent experiments.

Considering the pursuit of water detail extraction in remote sensing images, deep neural networks can obtain better performances. ResNet [38] solves the vanishing gradient problem in the backpropagation part of this deep network by introducing a shortcut connection that adds the output of the previous layers to the output of this layer and feeds the summed result to the activation function as the output. ResNet has several variants, such as ResNet50 and ResNet 101 (Table 2). However, according to a ResNet principle, the number of network layers can be deeper, and an increase layers may worsen or slightly improve the accuracy. Considering speed and accuracy, we chose ResNet101 as the backbone network for optimization.

To better extract water bodies in remote sensing images, this study improves on the ResNet101 network. ResNet101 outputs features of size $16 \times 16 \times 2048$, and upsampling with $1 \times 1$ convolution loses a lot of the boundary and semantic information. Therefore, the low-level features output from the first and second convolutional ResNet modules were combined with the high-level features upsampled by the SPM and MPM. Fig 10 shows the prediction results. The lower-level features contain the boundary information of large water bodies, and the accuracy of extracting water bodies is ensured by combining the training.

**Table 2. Comparison of various ResNet networks.**

| Layername | Output size | 18-layer | 34-layer | 50-layer | 101-layer | 152-layer |
|---|---|---|---|---|---|---|
| Conv1 | $112 \times 112$ | $7 \times 7$, 64, stride2 | | | | |
| Conv2_x | $56 \times 56$ | $3 \times 3$ maxpool, stride2 | | | | |
| | | $\begin{bmatrix} 3 \times 3 & 64 \\ 3 \times 3 & 64 \end{bmatrix} \times 2$ | $\begin{bmatrix} 3 \times 3 & 64 \\ 3 \times 3 & 64 \end{bmatrix} \times 3$ | $\begin{bmatrix} 1 \times 1 & 64 \\ 3 \times 3 & 64 \\ 1 \times 1 & 256 \end{bmatrix} \times 3$ | $\begin{bmatrix} 1 \times 1 & 64 \\ 3 \times 3 & 64 \\ 1 \times 1 & 256 \end{bmatrix} \times 3$ | $\begin{bmatrix} 1 \times 1 & 64 \\ 3 \times 3 & 64 \\ 1 \times 1 & 256 \end{bmatrix} \times 3$ |
| Conv3_x | $28 \times 28$ | $\begin{bmatrix} 3 \times 3 & 128 \\ 3 \times 3 & 128 \end{bmatrix} \times 2$ | $\begin{bmatrix} 3 \times 3 & 128 \\ 3 \times 3 & 128 \end{bmatrix} \times 4$ | $\begin{bmatrix} 1 \times 1 & 128 \\ 3 \times 3 & 128 \\ 1 \times 1 & 512 \end{bmatrix} \times 4$ | $\begin{bmatrix} 1 \times 1 & 128 \\ 3 \times 3 & 128 \\ 1 \times 1 & 512 \end{bmatrix} \times 4$ | $\begin{bmatrix} 1 \times 1 & 128 \\ 3 \times 3 & 128 \\ 1 \times 1 & 512 \end{bmatrix} \times 8$ |
| Conv4_x | $14 \times 14$ | $\begin{bmatrix} 3 \times 3 & 256 \\ 3 \times 3 & 256 \end{bmatrix} \times 2$ | $\begin{bmatrix} 3 \times 3 & 256 \\ 3 \times 3 & 256 \end{bmatrix} \times 6$ | $\begin{bmatrix} 1 \times 1 & 256 \\ 3 \times 3 & 256 \\ 1 \times 1 & 1024 \end{bmatrix} \times 6$ | $\begin{bmatrix} 1 \times 1 & 256 \\ 3 \times 3 & 256 \\ 1 \times 1 & 1024 \end{bmatrix} \times 23$ | $\begin{bmatrix} 1 \times 1 & 256 \\ 3 \times 3 & 256 \\ 1 \times 1 & 1024 \end{bmatrix} \times 28$ |
| Conv5_x | $7 \times 7$ | $\begin{bmatrix} 3 \times 3 & 512 \\ 3 \times 3 & 512 \end{bmatrix} \times 2$ | $\begin{bmatrix} 3 \times 3 & 512 \\ 3 \times 3 & 512 \end{bmatrix} \times 3$ | $\begin{bmatrix} 1 \times 1 & 512 \\ 3 \times 3 & 512 \\ 1 \times 1 & 2048 \end{bmatrix} \times 3$ | $\begin{bmatrix} 1 \times 1 & 512 \\ 3 \times 3 & 512 \\ 1 \times 1 & 2048 \end{bmatrix} \times 3$ | $\begin{bmatrix} 1 \times 1 & 512 \\ 3 \times 3 & 512 \\ 1 \times 1 & 2048 \end{bmatrix} \times 3$ |
| | $1 \times 1$ | Average pool, 1000-d fc, softmax | | | | |

## Evaluation index

To validate the effectiveness of the improved DeepLabv3+ network structure proposed, we used PA, mIOU, and recall as main evaluation metrics, where PA is the pixel accuracy, reflecting the ratio of pixels with correct prediction categories to the total pixels. The mIOU value is an important measure of image segmentation accuracy, which can be interpreted as the average crossover ratio, i.e., the IOU value calculated for each category. A higher mIOU value generally indicates better classification and prediction. The recall rate is used to calculate the ratio

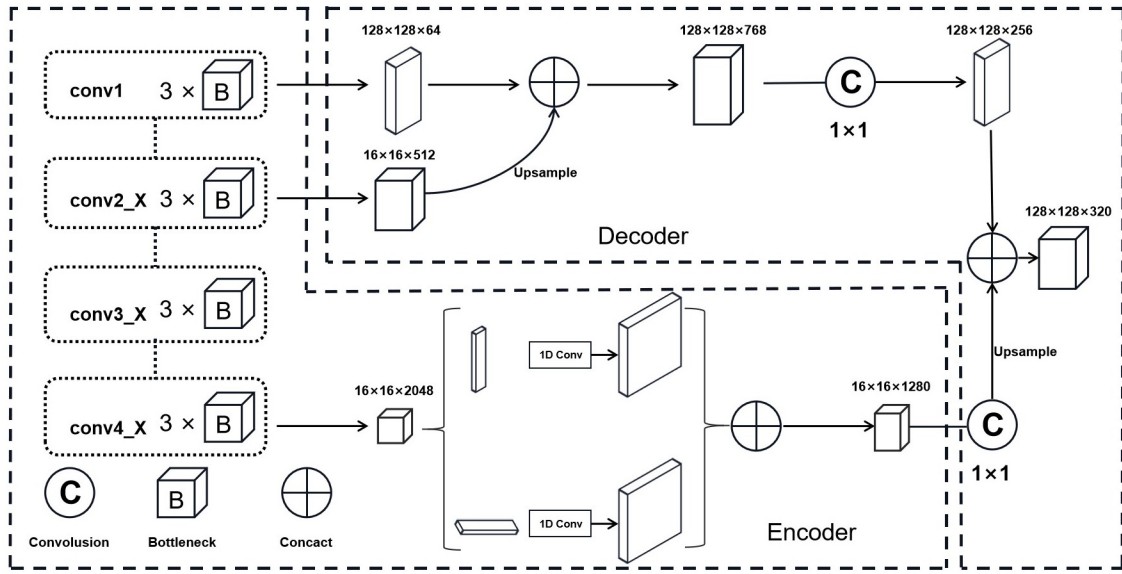

**Fig 10. Improved encoder-decoder model.**

of correctly classified water body pixels to the total number of pixels labeled as water bodies in the image. Its calculation formula can be given as Eqs (2), (3) and (4).

$$PA = \frac{T_P + T_N}{T_P + T_N + F_P + F_N},$$

(2)

$$mIOU = \frac{1}{k+1} \sum_{i=0}^{k} \frac{T_P}{T_P + F_P + F_N},$$

(3)

$$Recall = \frac{T_P}{T_P + F_N},$$

(4)

where $T_P$ represents the number of water pixels correctly classified, $T_N$ denotes the number of non-water pixels correctly classified, $F_P$ is the number of non-water pixels misclassified as water, and $F_N$ represents the number of water-body pixels misclassified as non-water bodies.

## Results

As shown in Fig 11, the overall trend of loss decreases as the number of training rounds increases. In the fiftieth training round, the loss function shows a sharp oscillation. We speculate that the reason for the oscillation is that a neuron in the network had a significant impact on the weights; thus, we added an additional dropout layer to the network. The trend of the loss after adding the dropout layer is shown on the right of Fig 11; evidently, the change in the loss tends to be smooth, and the convergence speed is accelerated.

Considering the different textures, shapes and colors of different bodies of water (e.g., lakes, river tributaries and main streams), the generalization ability of deep learning methods may be limited. To verify the effectiveness of our water body extraction method in different regions, in addition to the regions selected in Fig 3, we selected two representative regions, Chongqing and Chaozhou, China. Fig 12(a) and 12(b) show that the former contains a large basin, and the latter a large number of water bodies with multiple tributary structures. We applied the same false color and generative response network data enhancement to both images. Both images contain a large amount of information about small bodies of water, which is more complex than the training data.

To demonstrate the excellent performance of our method in extracting and distinguishing tiny water bodies, we chose locations with small watershed areas to compare against some of the models mentioned in the introduction. The data used corresponded to the training and test sets, and the test results are shown in Fig 13.

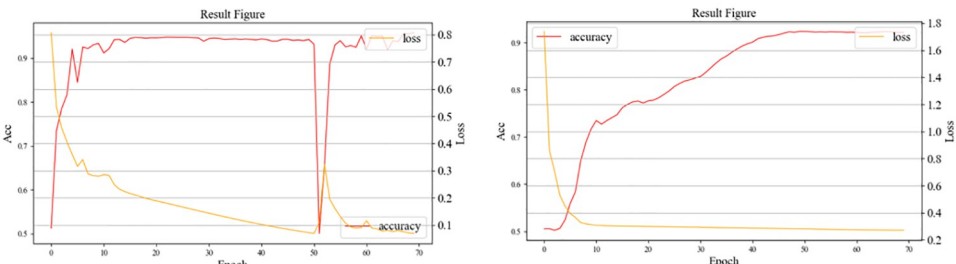

**Fig 11. Evolution of the loss functions during training.**

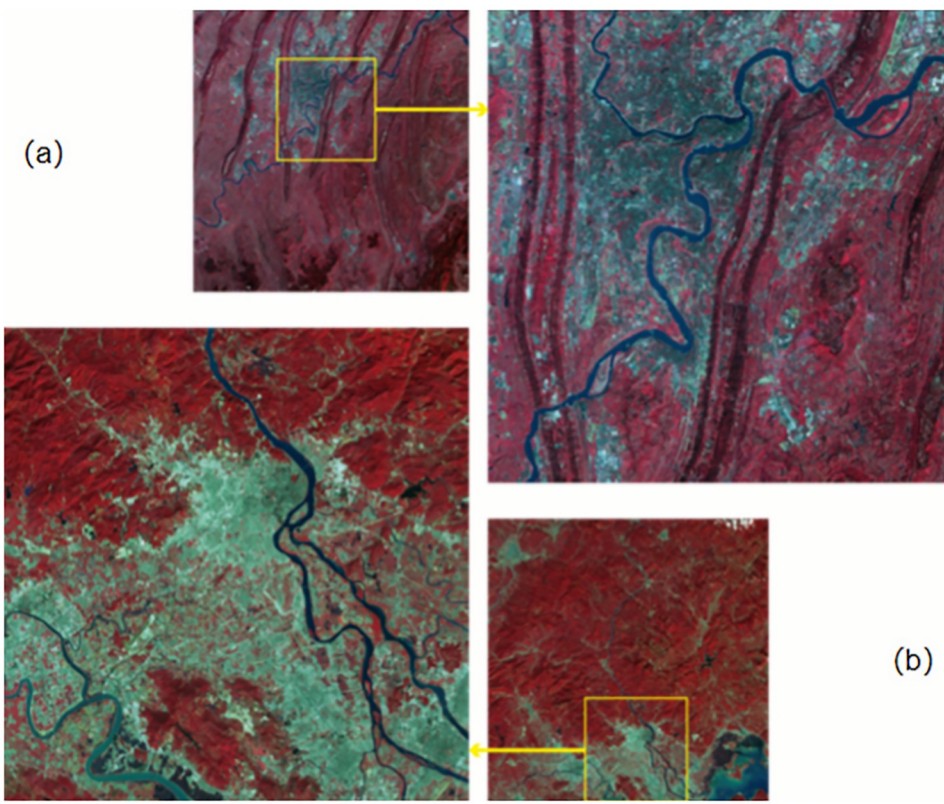

**Fig 12. Two remote sensing images from Chongqing (a) and Chaozhou (b) in China are selected as test data sources.** Reprinted from www.gscloud.cn under a CC BY license, with permission from Dr. Qinghui Lin, original copyright 2019.

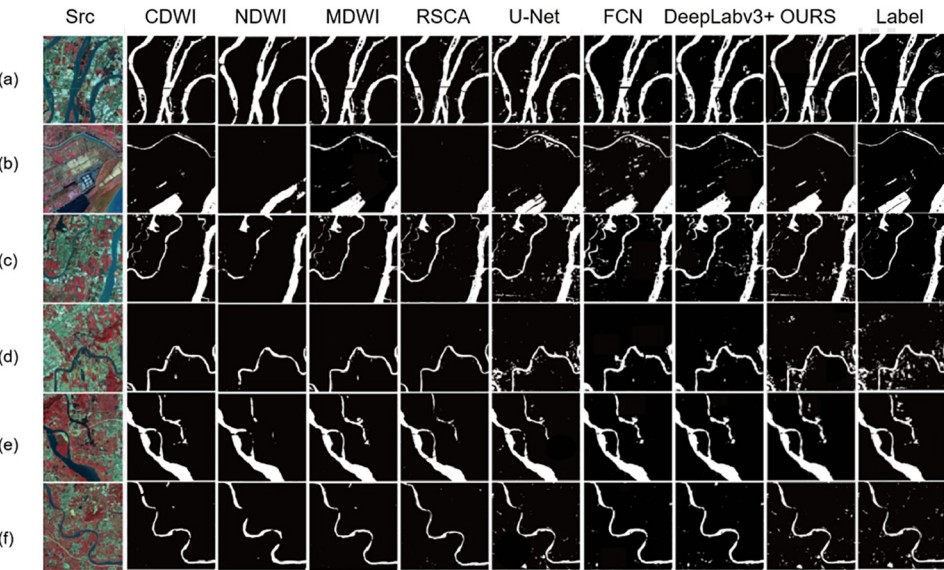

**Fig 13. Comparison of water body extraction results of different models.**

Rows (a)–(c) in Fig 13 were derived from the divided test area in Fig 4. From row (a) in Fig 13, our model can be determined to better extract the spatial information of remote sensing images than the other models in the very small river extraction task. For other objects in the water, our model has a higher accuracy and can effectively distinguish the water from the parts that are not water, although the existence of objects in the water is expected. Because of the stripe pooling, which is different from the traditional square pooling, the feature information extraction in the horizontal or vertical direction becomes freer. Row (b) in Fig 13 shows that the present model is better at suppressing noise points while extracting water bodies. When extracting small water, the band information may be fuzzy, and the traditional remote sensing extraction method is not good, which generally shows that small waters are not considered as water. In existing deep learning models, because small water is highly similar to the surrounding environment, such models often misclassify non-water parts as water. The GAN network adopted by us can well avoid these two extreme situations. The GAN network accentuates the differences between small water and the surrounding environment, making the deep learning model more capable of grasping the key points and correctly determine which parts are water. Row (c) in Fig 13 shows that our method obtains more complete and smoother edge details of the water body compared with the other methods. As remote sensing images are generally large, we had to cut and scale a complete remote sensing image to save computing resources, and the Lanczos algorithm we adopted ensured that the images did not affect the training results. From these results, we can observe that the existing semantic segmentation model can also extract water bodies, but it generates a large number of noisy data points during the extraction process, misclassifying non-water body parts as water and affecting the overall extraction effect. While the traditional water body extraction method can distinguish the water boundary well, it performs poorly in fine water body extraction.

To verify the applicability of the proposed model, a trained model was used to extract water from the test data pictures in Figs 12 and 13. It can be seen that our model improves significantly on the extraction of tiny water bodies. Moreover, the completeness and edge refinement ability of water body extraction using the proposed method outperforms several of the models compared. The model trained using the data in Fig 4 can also have good water extraction performance on other data, which proves that the model has good applicability and good performance on other data.

We calculated the prediction performance of various models. First, we randomly selected some remote sensing image regions that were not involved during training, as well as regions excluded in the training set in Fig 4. Then, we combined them into the final test set to obtain the final model performance pairs, as shown in Table 3 and Fig 13. Our model achieved 94.72%, 93.16% and 93.87% in PA, mIOU and recall, respectively, which are higher than other models. The proposed method was verified to improve the accuracy of water extraction from remote sensing images. We also show the test results of whether GAN was used. Note that the test accuracy increased after using GAN. The most obvious is the DeepLabv3+ model, which increased in accuracy by approximately 0.7.

To further demonstrate the effectiveness of the GAN, we compared the prediction plots of the original data with the processed data (Fig 14). Although the training process was difficult, significant gains were achieved. When similar deep neural networks are used for classification or prediction in certain domains (e.g., vegetation extraction and classification), adversarial networks can be constructed to further enhance data features. As shown in Fig 14, water in the original data is highly similar to the surrounding environment, which is difficult to distinguish even with the naked eye. With the help of a GAN, water is distinguished from the surrounding environment, and the water features are strengthened. This change is helpful for any deep learning model extracting water.

**Table 3. Comparison of models.**

| Model | GAN | PA(%) | mIOU(%) | Recall(%) |
|---|---|---|---|---|
| NDWI | × | 90.346 | 86.424 | 87.338 |
| CDWI | × | 91.526 | 90.426 | 92.446 |
| MDWI | × | 91.684 | 90.362 | 89.236 |
| RCSA | × | 89.523 | 87.632 | 90.165 |
| U-Net | × | 91.116 | 84.022 | 88.671 |
| FCN | × | 91.887 | 90.807 | 91.060 |
| DeepLabv3+ | × | 92.135 | 90.426 | 92.446 |
| OURS | × | 94.179 | 92.556 | 93.573 |
| NDWI | ✓ | 88.594 | 87.237 | 87.862 |
| RCSA | ✓ | 90.156 | 88.335 | 90.568 |
| DeepLabv3+ | ✓ | 92.802 | 91.137 | 92.634 |
| U-Net | ✓ | 91.972 | 84.756 | 90.589 |
| MDWI | ✓ | 92.153 | 87.438 | 90.546 |
| CDWI | ✓ | 91.837 | 89.434 | 91.582 |
| FCN | ✓ | 92.726 | 91.262 | 91.638 |
| OURS | ✓ | 94.723 | 93.167 | 93.874 |

As the edges are composed of gray-level jump points, which have a high spatial frequency, we used high-pass filtering to let the high-frequency components pass through smoothly while suppressing the low-frequency components. By enhancing the high-frequency component, the edges of the image can be sharpened, thereby achieving image sharpness. When the image is captured with under- or overexposure, the range of the image recording device is too narrow, and other factors can create insufficient contrast, making the details of the image indistinguishable. We transformed the gray-level of each pixel in this experiment to expand the range of image gray-level for image enhancement. To verify the effectiveness of the proposed data

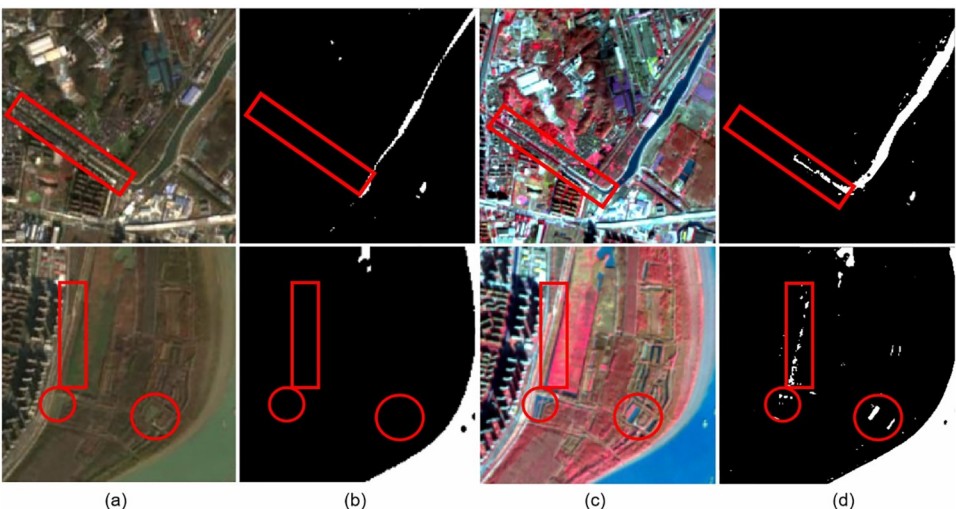

**Fig 14. Results of our proposed method on two selected samples from the testing dataset.** (**a**) Original remote sensing images containing tiny water bodies; (**b**) extraction results of existing methods; (**c**) generated remote sensing images; (**d**) extraction results of our proposed method. Reprinted from www.gscloud.cn under a CC BY license, with permission from Dr. Qinghui Lin, original copyright 2019.

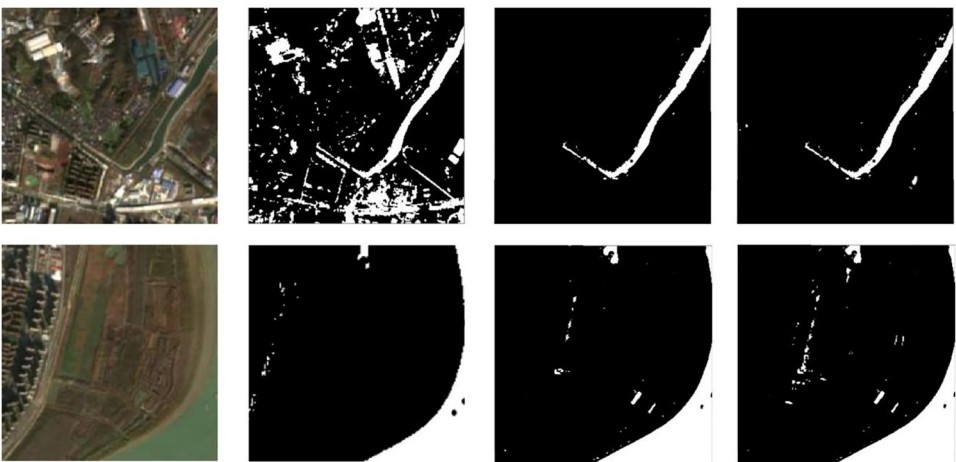

**Fig 15. The result of data processing comparison experiment.** From left to right, the original image, the result of grayscale transformation, the result of high-pass filtering and the result of this research method. Reprinted from www.gscloud.cn under a CC BY license, with permission from Dr. Qinghui Lin, original copyright 2019.

processing method, we conducted a data processing comparison experiment, and the results are shown in Fig 15.

From Fig 15, although the image after a simple grayscale transformation also has the ability to roughly identify parts of water bodies, the grayscale transformed image transforms every pixel of the image, which is less effective in identifying similar parts and cannot distinguish pixel information around water bodies well and performs poorly in urban water body extraction. The high-pass filtered image increases the distinction between high and low frequency components, and makes the edges of the body of water clearer to some extent. From the result figure, we can see that the image after adding high-pass filtering processing is the edge delineation of water bodies is clearer, but the extraction effect of small water bodies is poor. In addition, the proposed data processing method effectively realizes large water-body edge extraction as well as small water-body identification, demonstrating superiority through the recognition result map.

As covered and non-covered ground objects are similar in size, when the training neural network model, as the number of network training iterations increases, it can cause overfitting with the training data or non-convergence in the network. These problems are solved using single input scale and multi-scale features. By improving the structure of the input image, the interpolation algorithm is used to reduce the images of different scales into a uniform input, and the method of multiple input scales and scale features is used to extract the water bodies. The advantage of this method is the use of the interpolation algorithm to expand feature differences between neighboring pixels (Fig 16).

To ensure the accuracy of the models derived from the experiments, we reselected experimental data from the CIFAR, AI Challenger, and COCO datasets to validate the validity of this model. The CIFAR-10 dataset contains 60,000 color images of size $32 \times 32$, which we divided into 10 classes of 6,000 images each. We divided these 60,000 images into 50,000 training images and 10,000 test images. For training, we divided the dataset into five training batches and one test batch, each batch containing 10,000 images. The test batch contains 1,000 images randomly selected from each category. The remaining images appear in five sequential batches in a random order. Because it is random, the number of images of different classes contained in different batches may not be equal. AI Challenger contains 50,000 labeled images of 27 diseases of 10 plant species (i.e., apple, cherry, grape, citrus, peach, strawberry, tomato, pepper,

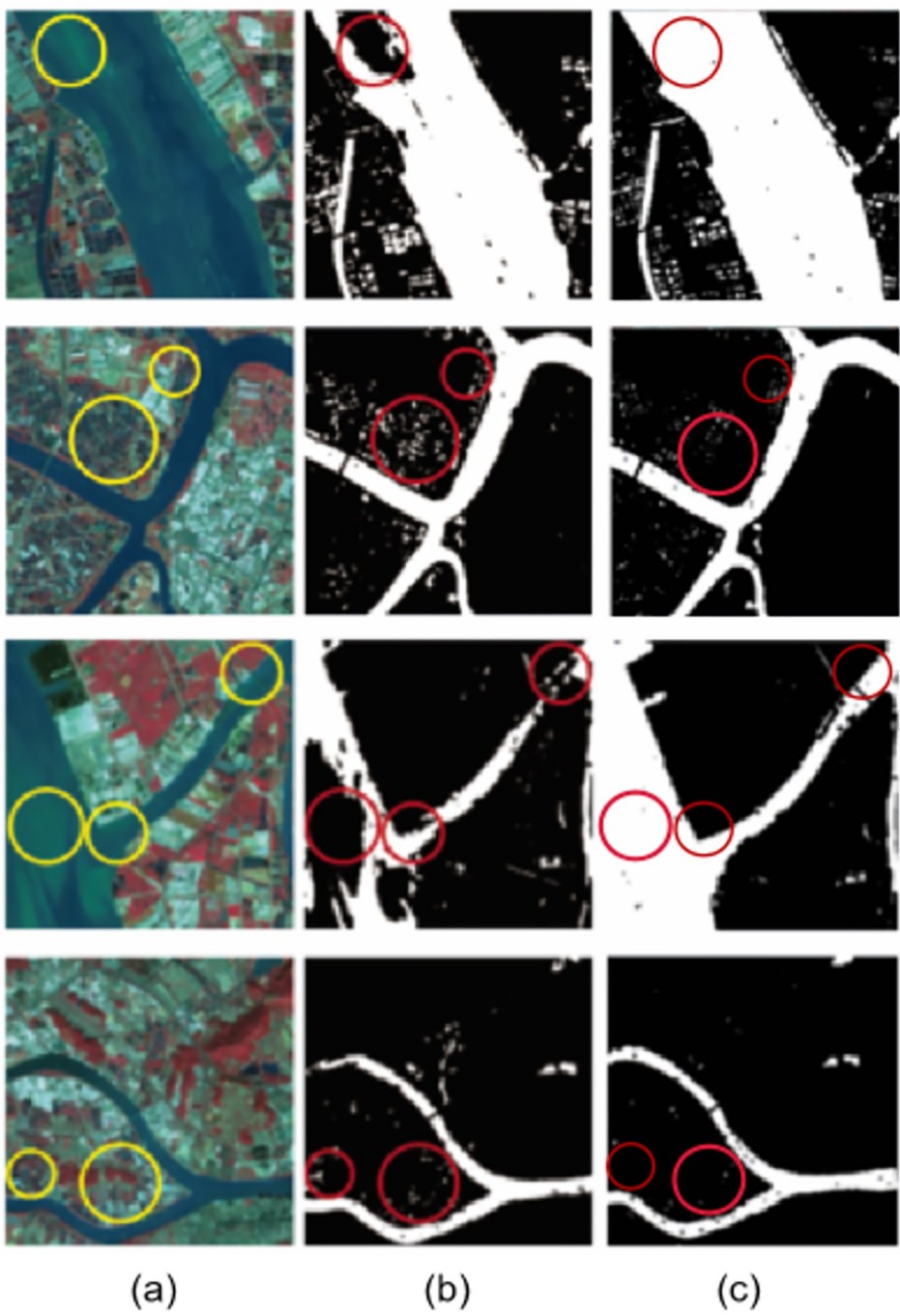

**Fig 16. Prediction results under different data processing methods.** Comparison of the prediction results for single input and multi-scale input. (**a**) is the original image, (**b**) is a single input, and (**c**) is a multi-scale input result. Reprinted from www.gscloud.cn under a CC BY license, with permission from Dr. Qinghui Lin, original copyright 2019.

corn, and potato) with 61 classifications. Here, we selected corn diseases with ten classifications for testing. The COCO dataset contains 1.5 million targets, 80 target object categories (pedestrians, cars, elephants, etc.), and 91 stuff categories (e.g., grass, walls, and sky). To verify the effectiveness of the proposed method, we only selected 10 data categories from them, such

**Table 4. Test performance of the research methods in this paper on three public datasets.**

| Data Category | Acc | Auc | Recall | Precision | F1 |
|---|---|---|---|---|---|
| CIFAR-10(10) | 0.9271 | 0.9837 | 0.9052 | 0.9080 | 0.9047 |
| COCO(10) | 0.9550 | 0.9376 | 0.8692 | 0.8995 | 0.8704 |
| AI Challenger(10) | 0.9212 | 0.8720 | 0.8512 | 0.8689 | 0.8586 |

as car and sky, for testing and comparison. The test results for the three public datasets are shown in Table 4.

As shown in Table 4, the proposed model still has excellent classification and recognition on public datasets, particularly in the COCO data classification task, with a recognition accuracy of 95.50%. By adding AI Challenger data, we still have a 92.12% reliability performance, proving that the data have minimal impact on the extraction effect of the network. The research method is proved to effectively extract feature information from the data through different experiments. It also significantly improves the accuracy of image classification tasks and has a strong compatibility with different data, exhibiting the robustness and capability of this research model.

## Discussion

Our experimental training data came from the Yangtze River Basin and Pearl River Delta in China, two regions with significantly different water clarity, microbial content, and water eutrophication levels. For traditional remote sensing extraction methods, a universal manner of obtaining water is lacking. The training set images contain numerous small branch rivers and farmlands. We expect to test the generalization ability of the model with representative data to demonstrate the effectiveness of our proposed method. This is also a challenge for the existing neural network model. Subsequent experiments above proved that our method is correct. By testing the model on data from a large number of water bodies in cities without prior training, the results show that our model can extract small ponds and rivers with high accuracy. Although this paper emphasizes the extraction of small water bodies in remote sensing images, the accuracy of the basic task of large water extraction was also demonstrated during various experiment stages. Comparing the results with and without a GAN, the accuracy with a GAN increases by approximately 0.6%, which is not a large increase, but proves the effectiveness of GANs in the feature enhancement of tiny water bodies. In addition, in comparing single and multi-scale inputs, the multi-scale input can better segment boundaries of water bodies, proving its necessity and effectiveness.

## Conclusion

This paper proposed a new water body extraction method for remote sensing images. The proposed method enhances the features of tiny water bodies in remote sensing images and replaces the original pooling method with strip pooling. In addition, the method provides a convenient multi-scale input strategy, and it comprises three stages. First, preprocessing was performed using false color processing, and remote sensing image reconstruction and enhancement were performed using GAN networks. Second, the training set was enriched with diversity on limited data, and a strategy was developed to achieve multi-scale input while lowering the training cost. Finally, the DeepLabv3+ network was improved using SPM and MPM modules, rather than the ASPP, to extract water bodies from satellite remote sensing images. Experiments show that, unlike existing methods ineffective in extracting tiny water

bodies and unable to distinguish water bodies from urban buildings, the proposed method is effective in extracting tiny water bodies and accurately classifies water bodies and urban buildings in large-scale remote sensing spaces. In a future study, we plan to extend the remote sensing image database to provide data support for future research and test various combinations of network modules, training strategies, and preprocessing schemes to further perfect the results. In addition to our method in terms of water extraction, the advantages of this method are promising in extracting remote sensing fields and providing a new way of thinking. In today's diversified world, combining different areas has also gradually become a trend of solving the problem.

## Supporting information

**S1 File. File containing code.** All codes used in this project are included in the S1_File.zip. (ZIP)

## Acknowledgments

Thanks to Editage for providing English language support.

## Author Contributions

**Data curation:** Jie Liao, Chengwu Zhang.

**Investigation:** Jie Liao, Chengwu Zhang.

**Methodology:** Yuanjiang Luo, Ao Feng, Xingqiang Zheng, Haibo Pu.

**Resources:** Hongxiang Li, Danyang Li, Xuan Wu, Xingqiang Zheng, Haibo Pu.

**Software:** Yuanjiang Luo, Hongxiang Li, Xuan Wu, Haibo Pu.

**Writing – original draft:** Ao Feng, Hongxiang Li.

**Writing – review & editing:** Yuanjiang Luo, Ao Feng.

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
