## [Decision Letter · Decision Letter 0]

19 May 2022

PONE-D-22-06183A New Deep Learning Method for Efficient Extraction of Small Water from Remote Sensing ImagesPLOS ONE

Dear Dr. Pu,

Thank you for submitting your manuscript to PLOS ONE. After careful consideration, we feel that it has merit but does not fully meet PLOS ONE’s publication criteria as it currently stands. Therefore, we invite you to submit a revised version of the manuscript that addresses the points raised during the review process.

The manuscript must be corrected in all points indicated by the reviewers, such as:

1) The abstract need to be slightly improved.

2) The English still needs a thorough revision.

3) Discuss all innovative points proposed in the study.

We look forward to receiving your revised manuscript.

Kind regards,

Claudionor Ribeiro da Silva

Academic Editor

PLOS ONE

Journal Requirements:

Whilst you may use any professional scientific editing service of your choice, PLOS has partnered with both American Journal Experts (AJE) and Editage to provide discounted services to PLOS authors. Both organizations have experience helping authors meet PLOS guidelines and can provide language editing, translation, manuscript formatting, and figure formatting to ensure your manuscript meets our submission guidelines. To take advantage of our partnership with AJE, visit the AJE website (http://aje.com/go/plos) for a 15% discount off AJE services. To take advantage of our partnership with Editage, visit the Editage website (www.editage.com) and enter referral code PLOSEDIT for a 15% discount off Editage services.  If the PLOS editorial team finds any language issues in text that either AJE or Editage has edited, the service provider will re-edit the text for free.

A clean copy of the edited manuscript (uploaded as the new *manuscript* file).

4. We note that Figures 4, 5, 6, 7, 9, 11 and 13 in your submission contain satellite images which may be copyrighted. All PLOS content is published under the Creative Commons Attribution License (CC BY 4.0), which means that the manuscript, images, and Supporting Information files will be freely available online, and any third party is permitted to access, download, copy, distribute, and use these materials in any way, even commercially, with proper attribution. For these reasons, we cannot publish previously copyrighted maps or satellite images created using proprietary data, such as Google software (Google Maps, Street View, and Earth). For more information, see our copyright guidelines: http://journals.plos.org/plosone/s/licenses-and-copyright.

a) You may seek permission from the original copyright holder of Figures 4, 5, 6, 7, 9, 11 and 13 to publish the content specifically under the CC BY 4.0 license.  

Reviewers' comments:

Reviewer's Responses to Questions

**Comments to the Author**

1. Is the manuscript technically sound, and do the data support the conclusions?

Reviewer #1: Yes

Reviewer #2: Yes

2. Has the statistical analysis been performed appropriately and rigorously? 

Reviewer #1: Yes

Reviewer #2: N/A

3. Have the authors made all data underlying the findings in their manuscript fully available?

Reviewer #1: No

Reviewer #2: Yes

4. Is the manuscript presented in an intelligible fashion and written in standard English?

Reviewer #1: Yes

Reviewer #2: Yes

5. Review Comments to the Author

Reviewer #1: The authors proposed aa new water body extraction method based on strip pooling for small water from remote sensing images. Generally speaking, this paper is well written and easy to follow, however, needs further revisions before publication. See below for detailed comments.

(1) The abstract need to be slightly improved.

(2) The use of deep learning methods for remote sensing intelligent processing should be mentioned in the Introduction Section, such as [1-4]:

[1] A novel water body extraction neural network (WBE-NN) for optical high-resolution multispectral imagery [J] Journal of Hydrology, doi: 10.1016/j.jhydrol.2020.125092

[2] Automatic mapping of urban green spaces using a geospatial neural network[J] GIScience & Remote Sensing, doi: 10.1080/15481603.2021.1933367.

[3] Deep Learning in Remote Sensing: A Comprehensive Review and List of Resources [J]. IEEE Geoscience and Remote Sensing Magazine, doi: 10.1109/MGRS.2017.2762307.

[4] Thick Clouds Removing From Multitemporal Landsat Images Using Spatiotemporal Neural Networks [J]. IEEE Transactions on Geoscience and Remote Sensing, doi: 10.1109/TGRS.2020.3043980.

(3) Please show the evolution of the loss functions during training at datasets.

(4) Justify your selection of bands at remote sensing images.

(5) The English still needs a thorough revision. I suggest the authors to have a native speaker correct the manuscript. Such as “Accurate acquisition of information on the distribution of surface water bodies is of

great significance in the fields of water resources investigation, comprehensive river management, water resources planning, flood and drought monitoring, and disaster assessment”

Reviewer #2: In this study, Luo et al. proposed a deep learning based water body extraction method for remote sensing images. The method consists of three main steps: 1. data processing step with false color and GAN processings for image enhancement; 2. multi-scale policies to enrich the limited training set; 3. an improved DeepLabv3+ model with the strip pooling was applied to extract the water bodies with different strip kernels. They trained and evaluated their approach on the inhouse sensing image and achieved the best performances compared to conventional and other deep learning methods.

Overall, this paper is technically sound and well-organized. It is mostly well written with good references.

The following are my comments to improve the manuscript:

1. The authors claimed, “We propose a strategy that enables multi-scale input while making the training cost lower.” Here, what does the training cost refer to, and why the multi-scale policy can reduce it?

2. Except for the very few remote sensing images used in the paper, it would be great if the authors could evaluate their method on some large public datasets for benchmarking and testing the generalization ability.

3. In the paper, the author proposed several innovative points to improve the final water body extraction performance, but only the data enhancement method with GAN processing was evaluated in the experiment section. It would be good to add more ablation studies to assess the proposed points empirically.

4. The paper needs some thorough proofreading. Here are a few examples:

a. The past and present tenses are often misused in the paper.

b. L112, a new -> ‘A new’

c. L 115, ‘In We introduced a bar pool by detailed qualitative...’ ?

d. L-287, ‘dilated convolution’

e. In Fig.5. Why the output of the discriminator network (after the sigmoid activation function) is a single image? It will make readers confused.

6. PLOS authors have the option to publish the peer review history of their article (what does this mean?). If published, this will include your full peer review and any attached files.

Reviewer #1: No

Reviewer #2: No

---

## [Author Response · Author response to Decision Letter 0]

2 Jul 2022

Reviewer#1, Concern # 1: The abstract need to be slightly improved.

Author response: We apologize for our vague description, the abstract section is a very important part and should be described clearly.

Author action: We have modified the abstract to make it read more smoothly.

In this study, false color processing and a generative adversarial network (GAN) were added to reconstruct the remote sensing images and enhance features of tiny water bodies. In addition, a multi-scale input strategy was designed to reduce the training cost. We input the processed data into a new water body extraction method based on strip pooling for remote sensing images, which is an improvement of DeepLabv3+. Strip pooling was introduced in the DeepLabv3+ network to better extract water bodies with a discrete distribution at long distances using different strip kernels.

Reviewer#1, Concern # 2: The use of deep learning methods for remote sensing intelligent processing should be mentioned in the Introduction Section, such as [1-4]:

[1] A novel water body extraction neural network (WBE-NN) for optical high-resolution multispectral imagery [J] Journal of Hydrology, doi: 10.1016/j.jhydrol.2020.125092

[2] Automatic mapping of urban green spaces using a geospatial neural network[J] GIScience & Remote Sensing, doi: 10.1080/15481603.2021.1933367.

[3] Deep Learning in Remote Sensing: A Comprehensive Review and List of Resources [J]. IEEE Geoscience and Remote Sensing Magazine, doi: 10.1109/MGRS.2017.2762307.

[4] Thick Clouds Removing From Multitemporal Landsat Images Using Spatiotemporal Neural Networks [J]. IEEE Transactions on Geoscience and Remote Sensing, doi: 10.1109/TGRS.2020.3043980.

Author response: We are grateful to you. These deep learning methods for intelligent processing of remote sensing provide more innovative and supportive references for this manuscript and make it more reasonable.

Author action: We have reorganized the introduction section and added these deep learning methods for remote sensing intelligent processing in it(lines 50 to 62).

With the development of artificial intelligence technology, applying deep learning to information extraction in the remote sensing field has become a hot topic for researchers. Some researchers have applied semantic segmentation to remote sensing image interpretation and achieved good results [12, 13], such as automatic mapping method of urban green spaces(UGS) [14] and novel spatiotemporal neural network [15]. Recently, deep learning has been increasingly applied to the extraction of water body information from remote sensing images. Qi et al. combined convolutional neural network (CNN) with Markov model and used semi-supervised learning strategy to reduce data dependency improving the extraction performance of global and local water bodies by 7-10% [16]. Chen et al. developed a global spatial-spectral convolution and surface water body boundary refinement module to enhance surface water body features. They also designed the WEB-NN architecture to segment high-resolution remote sensing images [17].

Reviewer#1, Concern # 3: Please show the evolution of the loss functions during training at datasets.

Author response: We apologize for our oversight. After your reminder, the result of this article is much more convincing.

Author action: We redid the experiments and added the results in the results section. We made an analysis of the experimental results describing the evolution of the loss function during the training of the datasets(lines 353 to 359).

As shown in Fig11, the overall trend of loss decreases as the number of training rounds increases. In the fiftieth training round, the loss function shows a sharp oscillation. We speculate that the reason for the oscillation is that a neuron in the network had a significant impact on the weights; thus, we added an additional dropout layer to the network. The trend of the loss after adding the dropout layer is shown on the right of Fig11; evidently, the change in the loss tends to be smooth, and the convergence speed is accelerated.

Reviewer#1, Concern # 4: Justify your selection of bands at remote sensing images.

Author response: We are sorry for the poor expression in the previous version, which makes the article read vague. Actually, the NIR band used in this article is located in the highly reflective vegetation zones, reflecting plant information, and in the strong absorption zones of water bodies, enabling the identification of water-related geological formations and outlining water body boundaries.

Author action: We have added detailed description in the data processing section(lines 221 to 230).

In summary, we adopted the standard pseudo-color processing scheme involving the assignment of the green, red, and infrared bands of remote sensing data to the blue, green, and red bands of RGB, respectively. The NIR band used is located in the highly reflective vegetation zones, reflecting plant information, and in the strong absorption zones of water bodies, enabling the identification of water-related geological formations and outlining water body boundaries. The green and red bands further highlight the distinctions between water and vegetation and help improve the accuracy of water extraction. The experiment results show that the overall detection effect improved using standard false color processing.

Reviewer#1, Concern # 5: The English still needs a thorough revision. I suggest the authors to have a native speaker correct the manuscript. Such as “Accurate acquisition of information on the distribution of surface water bodies is of great significance in the fields of water resources investigation, comprehensive river management, water resources planning, flood and drought monitoring, and disaster assessment”

Author response: We are very sorry that we made such a low-level mistake. We have sent the paper to Editage Agency for language polishing.

Author action: We modified the part in the introduction section that you mentioned(lines 1 to 5).

Reviewer#2, Concern # 1: The authors claimed, “We propose a strategy that enables multi-scale input while making the training cost lower.” Here, what does the training cost refer to, and why the multi-scale policy can reduce it?

Author response: We are very sorry for our vague description and we have revised the relevant content in the original manuscript to make it read more clearly and understandable.

Author action: We have added detailed description of the problem in the input processing section(lines 267 to 279).

A larger input size of the input loses some image detail information, whereas a smaller input size generates a large amount of error owing to the complexity of the information contained in the remote sensing image which affects the final accuracy of the model. Both of these factors negatively affect the accuracy of the model to different degrees. The Sentinel-2A satellite remote sensing image data acquired in this study had a spatial resolution of 10m, but a river in the city is narrow and can be as small as 1 pixel wide in the image perception, and the width of the river crossing the city is much larger.The problem of extracting large water bodies while considering narrow rivers must be solved. The multi-scale input can train the model to accurately extract spatial information from images of different sizes, accounting for both local and global information to achieve good results in extracting large-area waters and small water bodies. Our multi-scale strategy reduces the training time cost while achieving the same training results.

Reviewer#2, Concern # 2: Except for the very few remote sensing images used in the paper, it would be great if the authors could evaluate their method on some large public datasets for benchmarking and testing the generalization ability.

Author response: We note that your reminder is very important, so we evaluated our approach on the CIFAR, AI Challenger, and COCO datasets to demonstrate the generalization ability of our method.

Author action: We add the results of our experiments in the result section(lines458 to 482).

To ensure the accuracy of the models derived from the experiments, we reselected experimental data from the CIFAR, AI Challenger, and COCO datasets to validate the validity of this model. The CIFAR-10 dataset contains 60,000 color images of size 32 × 32, which we divided into 10 classes of 6,000 images each. We divided these 60,000 images into 50,000 training images and 10,000 test images. For training, we divided the dataset into five training batches and one test batch, each batch containing 10,000 images. The test batch contains 1,000 images randomly selected from each category. The remaining images appear in five sequential batches in a random order. Because it is random, the number of images of different classes contained in different batches may not be equal. AI Challenger contains 50,000 labeled images of 27 diseases of 10 plant species (i.e., apple, cherry, grape, citrus, peach, strawberry, tomato, pepper, corn, and potato) with 61 classifications. Here, we selected corn diseases with ten classifications for testing. The COCO dataset contains 1.5 million targets, 80 target object categories (pedestrians, cars, elephants, etc.), and 91 stuff categories (e.g., grass, walls, and sky). To verify the effectiveness of the proposed method, we only selected 10 data categories from them, such as car and sky, for testing and comparison. The test results for the three public datasets are shown in Table4.

Table 4. Test performance of the research methods in this paper on three public datasets.

Data Category Acc Auc Recall Precision F1

CIFAR-10(10) 0.9271 0.9837 0.9052 0.9080 0.9047

COCO(10) 0.9550 0.9376 0.8692 0.8995 0.8704

AI Challenger(10) 0.9212 0.8720 0.8512 0.8689 0.8586

As shown in Table4, the proposed model still has excellent classification and recognition on public datasets, particularly in the COCO data classification task, with a recognition accuracy of 95.50%. By adding AI Challenger data, we still have a 92.12% reliability performance, proving that the data have minimal impact on the extraction effect of the network. The research method is proved to effectively extract feature information from the data through different experiments. It also significantly improves the accuracy of image classification tasks and has a strong compatibility with different data, exhibiting the robustness and capability of this research model.

Reviewer#2, Concern # 3: In the paper, the author proposed several innovative points to improve the final water body extraction performance, but only the data enhancement method with GAN processing was evaluated in the experiment section. It would be good to add more ablation studies to assess the proposed points empirically.

Author response: We apologize for our oversight. After your reminder, we added additional data ablation experiments to validate our proposed method.

Author action: We have added our experimental results in the results section(lines 427 to 448).

As the edges are composed of gray-level jump points, which have a high spatial frequency, we used high-pass filtering to let the high-frequency components pass through smoothly while suppressing the low-frequency components. By enhancing the high-frequency component, the edges of the image can be sharpened, thereby achieving image sharpness. When the image is captured with under- or overexposure, the range of the image recording device is too narrow, and other factors can create insufficient contrast, making the details of the image indistinguishable. We transformed the gray-level of each pixel in this experiment to expand the range of image gray-level for image enhancement. To verify the effectiveness of the proposed data processing method, we conducted a data processing comparison experiment, and the results are shown in Fig15.

From Fig15,although the image after a simple grayscale transformation also has the ability to roughly identify parts of water bodies, the grayscale transformed image transforms every pixel of the image, which is less effective in identifying similar parts and cannot distinguish pixel information around water bodies well and performs poorly in urban water body extraction. The high-pass filtered image increases the distinction between high and low frequency components, and makes the edges of the body of water clearer to some extent. From the result figure, we can see that the image after adding high-pass filtering processing is the edge delineation of water bodies is clearer, but the extraction effect of small water bodies is poor. In addition, the proposed data processing method effectively realizes large water-body edge extraction as well as small water-body identification, demonstrating superiority through the recognition result map.

Reviewer#2, Concern # 4: The paper needs some thorough proofreading. Here are a few examples:

a. The past and present tenses are often misused in the paper.

b. L112, a new -> ‘A new’

c. L 115, ‘In We introduced a bar pool by detailed qualitative...’ ?

d. L-287, ‘dilated convolution’

e. In Fig.5. Why the output of the discriminator network (after the sigmoid activation function) is a single image? It will make readers confused.

Author action: We are very sorry for these low-level mistakes we made.For the first to fourth points you raised, we have sent our paper to Editage Agency for language polishing. As for the fifth point you raised, we are sorry that the original image did not clearly represent the process of the network. We have reworked the flowchart for generating images via GAN network and added description of it(lines 237 to 253).

As the initial data in the original GAN is random noise and the network only requires the generated image of the generator to approximate the real image without setting constraints on its content, the generated image may not match our expected content despite its realism. To make the generated images fit our expected content as much as possible, we used two GANs in a cyclic manner to form the network, whose structure is shown in Fig5.In our GAN, we input the original image into the first GAN, use its generator G1 to generate images, and subsequently input the generated images into its discriminator D1 to discriminate whether the generated image of G1 is true according to the label. Then, the generated image is fed to generator G2 of the second GAN, and the generated image of G2 is given to the discriminator D2 of the second GAN to discriminate whether it approximates the original input image. In this manner, we obtain a generated image that is realistic to the label and retains the content of the original input image, enhancing the fine water body features in the image. In addition, to converge the imbalance between the generator and discriminator, we added artificial noise data to the output images of generator G1 . Fig 6 shows the original, the falsely colored and processed, and GAN-enhanced images.

---

## [Decision Letter · Decision Letter 1]

18 Jul 2022

New deep learning method for efficient extraction of small water from remote sensing images.

PONE-D-22-06183R1

Dear Dr. Pu,

We’re pleased to inform you that your manuscript has been judged scientifically suitable for publication and will be formally accepted for publication once it meets all outstanding technical requirements.

Kind regards,

Claudionor Ribeiro da Silva

Academic Editor

PLOS ONE

Additional Editor Comments (optional):

Reviewers' comments:

Reviewer's Responses to Questions

**Comments to the Author**

1. If the authors have adequately addressed your comments raised in a previous round of review and you feel that this manuscript is now acceptable for publication, you may indicate that here to bypass the “Comments to the Author” section, enter your conflict of interest statement in the “Confidential to Editor” section, and submit your "Accept" recommendation.

Reviewer #1: All comments have been addressed

Reviewer #2: All comments have been addressed

2. Is the manuscript technically sound, and do the data support the conclusions?

Reviewer #1: Yes

Reviewer #2: Yes

3. Has the statistical analysis been performed appropriately and rigorously? 

Reviewer #1: Yes

Reviewer #2: N/A

4. Have the authors made all data underlying the findings in their manuscript fully available?

Reviewer #1: Yes

Reviewer #2: Yes

5. Is the manuscript presented in an intelligible fashion and written in standard English?

Reviewer #1: Yes

Reviewer #2: Yes

6. Review Comments to the Author

Reviewer #1: The authors have given clearly answers to the questions. Therefore, I suggest accepting this manuscript.

Reviewer #2: I appreciate the authors’ responses and revision. My comments and concerns have been addressed appropriately. The revised manuscript can be considered for publication.

7. PLOS authors have the option to publish the peer review history of their article (what does this mean?). If published, this will include your full peer review and any attached files.

Reviewer #1: No

Reviewer #2: No

---

## [Editor Report · Acceptance letter]

27 Jul 2022

PONE-D-22-06183R1 

New deep learning method for efficient extraction of small water from remote sensing images 

Dear Dr. Pu:

I'm pleased to inform you that your manuscript has been deemed suitable for publication in PLOS ONE. Congratulations! Your manuscript is now with our production department. 

Kind regards, 

on behalf of

Dr. Claudionor Ribeiro da Silva 

Academic Editor

PLOS ONE